# Digital Tomosynthesis as a Problem-Solving Technique to Confirm or Exclude Pulmonary Lesions in Hidden Areas of the Chest

**DOI:** 10.3390/diagnostics13061010

**Published:** 2023-03-07

**Authors:** Elisa Baratella, Emilio Quaia, Filippo Crimì, Pierluca Minelli, Vincenzo Cioffi, Barbara Ruaro, Maria Assunta Cova

**Affiliations:** 1Institute of Radiology, Department of Medical Surgical and Health Sciences, Cattinara Hospital, University of Trieste, 34149 Trieste, Italy; 2Institute of Radiology, Department of Medicine-DIMED, University of Padova, 35128 Padova, Italy; 3Pulmonology Unit, Department of Medical Surgical and Health Sciences, University Hospital of Cattinara, University of Trieste, 34149 Trieste, Italy

**Keywords:** chest digital tomosynthesis, pulmonary hidden areas, pulmonary lesions

## Abstract

Objectives: To evaluate the capability of digital tomosynthesis (DTS) to characterize suspected pulmonary lesions in the so-called hidden areas at chest X-ray (CXR). Materials and Methods: Among 726 patients with suspected pulmonary lesions at CXR who underwent DTS, 353 patients (201 males, 152 females; age 71.5 ± 10.4 years) revealed suspected pulmonary lesions in the apical, hilar, retrocardiac, or paradiaphragmatic lung zones and were retrospectively included. Two readers analyzed CXR and DTS images and provided a confidence score: 1 or 2 = definitely or probably benign pulmonary or extra-pulmonary lesion, or pulmonary pseudo-lesion deserving no further diagnostic work-up; 3 = indeterminate lesion; 4 or 5 = probably or definitely pulmonary lesion deserving further diagnostic work-up by CT. The nature of DTS findings was proven by CT (*n* = 108) or CXR during follow-up (*n* = 245). Results: In 62/353 patients the suspected lung lesions were located in the lung apex, in 92/353 in the hilar region, in 59/353 in the retrocardiac region, and in 140/353 in the paradiaphragmatic region. DTS correctly characterized the CXR findings as benign pulmonary or extrapulmonary lesion (score 1 or 2) in 43/62 patients (69%) in the lung apex region, in 56/92 (61%) in the pulmonary hilar region, in 40/59 (67%) in the retrocardiac region, and in 106/140 (76%) in the paradiaphragmatic region, while correctly recommending CT in the remaining cases due to the presence of true solid pulmonary lesion, with the exception of 22 false negative findings (60 false positive findings). DTS showed a significantly (*p* < 0.05) increased sensitivity, specificity, and overall diagnostic accuracy and area under ROC curve compared to CXR alone. Conclusions: DTS allowed confirmation or exclusion of the presence of true pulmonary lesions in the hidden areas of the chest.

## 1. Introduction

Chest X-ray (CXR) remains the most commonly performed radiologic examination in clinical practice. However, doubtful or equivocal findings due to pulmonary, extra-pulmonary, or pseudo-lesions are often reported and small lesions may be missed [1,2]. This is particularly evident in the so-called “hidden areas” of the chest, including the apical, hilar, retrocardiac, and paradiaphragmatic lung zones, where lesions may have poor conspicuity for the presence of overlying anatomical structures (anatomical noise) [3,4]. Numerous methods have been proposed to address perceptual limitations in CXRs, including dual-energy chest radiography, bone subtraction, and computer-aided diagnosis [5]. Digital chest tomosynthesis (DTS) is a technique providing some of the tomographic advantages of CT but at lower costs and radiation dose [6,7,8,9,10,11,12,13,14]. During DTS, an X-ray tube moves along the patient, acquiring a series of X-ray projections at different angles and each at very low dose; this set of images is then used in a reconstruction algorithm similar to CT (Filtered Back Projection). By varying the amount of shift, planar images at different depths can be reconstructed and objects outside of the focus plane are rendered with a varying amount of blur. This reconstruction results in a stack of images representing a set of image planes through the anatomy and parallel to the flat panel detector. Different from linear tomography, in DTS, a single sweep of the tube results in a complete set of plane images that cover the entire anatomical depth. The major advantages of DTS over conventional CXR are the removal of overlying anatomical structures, the enhancement of local tissue separation, and availability of depth information of the structure of interest [7,8,15]. Therefore, DTS is expected to have an increased sensitivity compared to CXR in detecting small nodules and nodules in those regions of the chest where the anatomical noise reduces the contrast difference between the lesion and the surrounding area. Previous studies have shown that DTS increases pulmonary lesion conspicuity, improving diagnostic accuracy and confidence in confirming or ruling out lesions suspected on CXR and increasing sensitivity in detecting CT-proven lung nodules [9,10,11,12,13,14,16,17]. However, there are few studies evaluating the diagnostic performance of DTS in detecting pulmonary nodules based on their location, particularly in those regions where the superimposition of structures and complexity of the area render the assessment on chest X-rays challenging. The aim of this study was to evaluate the capability of digital tomosynthesis (DTS) to characterize suspicious pulmonary lesions in these so-called “hidden areas” at chest X-ray (CXR).

## 2. Materials and Methods

This was a single-center retrospective study, approved by the ethics committee of our hospital, with informed consent obtained from all patients.

### 2.1. Patients Population

Patients who revealed suspected pulmonary lesion(s) on CXR and underwent DTS from June 2009 to June 2018 were identified. We considered eligible for the present study all patients with suspected pulmonary lesion(s), appearing as areas of increased opacity or pulmonary nodules that could not reliably be considered present or located within the so-called “hidden areas” of the lung—base, apex, hilar retrocardiac, and paradiaphragmatic areas—on CXR. Inclusion criteria were DTS performed within 15 days from CXR and absence of respiratory artifacts on DTS preventing correct image assessment.

Figure 1 summarizes the study population accrual according to enhancing the quality and transparency of health research guidelines.

### 2.2. Acquisition Protocols

CXR examinations were obtained with a computed radiography (Kodak DirectView CR 975; Carestream, Rochester, MN, USA) or a digital radiography (Definium 8000; GE Healthcare, Chalfont St Giles, UK) system. X-ray images were acquired in the posteroanterior and left lateral views at the wall stand with a focal spot size of 0.6 mm, and a stationary antiscatter grid (70 lines per cm; ratio 13:1). DTS images (Definium 8000; GE Healthcare, Chalfont St Giles, UK) were acquired with a single linear sweep of the X-ray tube over an angle of ±15°, with 61 low-dose projections acquired at regular angular intervals during the tube sweep. Chest CT was performed with a 256-row multi-detector CT system (Brilliance iCT 256, Philips, Best, The Netherlands) and consisted of an unenhanced CT or CT scan acquired after the intravenous bolus injection of iodinated contrast material. The scanning parameters were: rotation time, 0.5 s; beam collimation, 128 × 0.625 mm; normalized pitch, 0.993; section reconstruction thickness, 1 mm; tube voltage,120 kVp; tube current (mA), 229; and field of view, 35 cm.

### 2.3. Image Analysis

Visual analysis of CXR and DTS images of each patient was carried out by two radiologists (E.B., V.C.) with an experience of 10 and 15 years in thoracic imaging, respectively, and who were not involved in the preliminary image interpretation for patient care. Readers were allowed to scroll the DTS images and use processing tools such as brightness and image contrast adjustment or magnification. All readings were performed on a picture archiving and communications system (PACS)-integrated workstation (19-inch TFT display, resolution 2560 × 1600 pixels) at a central location. First the CXR images, and then the DTS images of each patient were examined consecutively in the same reading session by both readers, who worked in consensus and were blinded on patient identification and clinical history. Discrepant interpretations (*n* = 10 patients) were resolved by consensus through the involvement of an additional reader with similar experience in thoracic imaging (E.Q.). The readers expressed a diagnostic confidence score for each lesion: (1) definitely or (2) probably benign pulmonary lesions or extra-pulmonary lesion, that is a lesion not contained in the limits of lung parenchyma or a pseudolesion due to vascular kinking, pleural lesion, or overlap of vascular and bone structures of the thoracic wall; (3) indeterminate, for a doubtful lesion nature; (4) probably; or (5) definitely malignant pulmonary lesion, that is a lesion contained in the anatomic limits of lung parenchyma. Then, both readers were asked to identify the same lesions on DTS images without a washout time, and to express diagnostic confidence according to the same scoring system employed for CXR.

### 2.4. Diagnostic Workup

Each patient underwent imaging follow-up for an overall time of at least two years. The nature of DTS findings was proven by CT or CXR based on the preliminary image assessment. CT was performed if DTS revealed an indeterminate or solid lesion (score 3, 4 or 5). CT images at standard lung window settings (window level of −600 HU and window width of 2000 HU) were analyzed immediately after image acquisition by a consensus of two senior radiologists with 8 and 15 years of experience in chest imaging, respectively, who were not involved in the visual interpretation of CXR or DTS images. All lesions presenting overt malignant features at CT (irregular or spiculated margins, pleural, or vascular infiltrations) underwent surgical resection; the remaining lesions were characterized by CT follow-up performed no less than 6 months apart for at least 2 years, and CT-guided biopsy was performed on those lesions which showed progressive volume increase (a doubling time of less than 500 days was considered as indicative of a malignant lesion [18]). All lesions were considered benign if they contained fat or were calcified, disappeared during imaging follow-up, or decreased/remained unequivocally stable in size during serial examinations. Conversely, patients underwent a CXR follow-up at a mean of 6 months after DTS (range 3–8 months) if imaging findings revealed an overt benign pulmonary or extra-pulmonary nature or a pseudolesion (score 1–2). DTS imaging findings were considered proven when the lesion disappeared during imaging follow-up, CXR did not confirm any pulmonary lesion or it confirmed the presence of an overt benign pulmonary or extra-pulmonary lesion in the same region where readers visualized a suspected pulmonary lesion on DTS.

### 2.5. Statistical Analysis

Statistical analysis was performed using MedCalc for Windows version 11.2.1.0 (MedCalc Software, Mariakerke, Belgium). A per-patient analysis was performed with the marker lesion considered for the calculation of sensitivity, specificity, positive predictive value, and negative predictive value and overall diagnostic accuracy.

To assess the improvement in observers’ performance in correctly diagnosing pulmonary lesions, Chi-square test with Yates correction was employed [19]. The improvement in diagnostic confidence was assessed by receiver operating characteristic (ROC) curve analysis, and the method proposed by Hanley and McNeil was employed to compare the areas under each ROC curve [20,21].

The reference standard was the definitive diagnosis made by CT or follow-up in the patients that showed benign or suspect finding at DTS.

Patients that did not show any finding at DTS after CXR were excluded from the accuracy analysis.

The result of the DTS or CXR was considered as true-positive if the lesion was correctly assessed as a non-calcified pulmonary lesion (confidence score 4 or 5) or a lesion appearing as a parenchymal or ground-glass opacity, or as a solid or sub-solid ground-glass pulmonary nodule. A false positive was a benign pulmonary or extra-pulmonary lesion, or a pulmonary pseudolesion at reference standard incorrectly assessed as a lung lesion deserving further evaluation by CT (scores 4 or 5) or assessed as indeterminate (score 3). A true negative was a lesion correctly assessed as a benign pulmonary lesion (centrally calcified lesion, lesion with gross calcifications or calcified fibrotic scars with pulmonary architectural distortion), as an extra-pulmonary lesion, or as a pulmonary pseudolesion (scores 1, 2). A false-negative finding at DTS or CXR was a pulmonary lesion incorrectly assessed as a benign pulmonary or extra-pulmonary lesion, or as a pulmonary pseudolesion (score of 1 or 2) or assessed as indeterminate (score 3).

For all statistical tests, a *p* value <0.05 was considered to indicate a statistically significant difference. For analysis of the estimated dose, data were analyzed by a PC-based X-ray Monte Carlo program, PCXMC 2.0 (Radiation and Nuclear Safety Agency, Helsinki, Finland) [22]. The mathematical phantom used in PCXMC 2.0 for the adult phantom is based on the model specified by Cristy and Eckerman [23]. We employed the method previously described to calculate the effective dose for the postero-anterior and left lateral projections of CXR and for each DTS projection view [24]. The CT effective dose estimate was determined by using dose length product (DLP) measurements and appropriate normalized coefficients found in the European guidelines [25] for chest CT (0.017 mSv × mGy^−1^ × cm^−1^).

## 3. Results

Among 726 patients with suspected pulmonary lesions on CXR that underwent DTS, 353 patients with suspected lesions in hidden areas were retrospectively included (201 males, 152 females; age, 71.5 ± 10.4 years). In 62/353 (17%) patients, the suspected lung lesions were located in the lung apex, in 92/353 (26%) in the hilar region, in 59/353 (17%) in the retrocardiac region, and in 140/353 (40%) in the paradiaphragmatic region.

A total of 237 pulmonary or extrapulmonary alterations were identified with DTS: in 132/237 (56%) patients, a CT was performed, while in the remaining 105/237 patients (44%), CXR doubtful findings were resolved by DTS according to the preliminary image assessment provided by the radiologists. In eight patients, additional lung nodules (>5 mm and <1 cm in diameter) missed at CXR were detected at DTS and confirmed at CT (Figure 2).

During follow-up, the nature of DTS findings was proven by CT in 108 patients and by CXR in 245 patients. Among 77 lung opacities, there were 32 non-tumoral lung opacities, 5 squamous lung carcinomas, 28 pulmonary benign nodules, 5 lung adenocarcinomas, and 7 pulmonary metastases. The squamous lung carcinomas (diameter, 2–3 cm) appeared as a hilar enhancing mass invading bronchial or pulmonary vessel wall (*n* = 4) or as a peripheral enhancing mass invading the pleura (*n* = 1) and underwent surgical resection. The lung adenocarcinomas (diameter, 0.5–1.5 cm) appeared as solid non-calcified nodules with irregular or spiculated margins, pleural or vascular infiltrations at CT and underwent surgical resection. The lung metastases (diameter 7 mm–12 mm) were either solitary (*n* = 4) or multiple (*n* = 3) and appeared as solid nodules proven to be metastatic by CT-guided biopsy. Among 26 pulmonary scars, there were 25 benign pulmonary scars and 1 lung adenocarcinoma (diameter, 15 mm). The last lesion was proven by CT-guided biopsy after a significant volume increase in the lesion on follow-up CTs. Pleural plaques (*n* = 12) were located in the anterior or posterior thoracic wall. Pseudolesions included composite areas of increased opacity from bone focal sclerosis (Figure 3) or overlap of vascular and bone structures of the chest (*n* = 92 patients), vascular kinkings or ectasias (*n* = 15), prominent cardiac auricula or mediastinal profiles (*n* = 10), or anatomical lung variants such as accessory fissures (*n* = 5).

Table 1 reports the radiologic imaging findings on DTS of all CXR findings included in the analysis, and the final diagnosis.

DTS correctly characterized the CXR findings as benign pulmonary or extrapulmonary (score 1 or 2) in 43/62 patients (69%) in lung apical region, in 56/92 (61%) in pulmonary hilar region, in 40/59 (67%) in retrocardiac region, and in 106/140 (76%) in the paradiaphragmatic region. DTS correctly recommended CT in the remaining cases for the presence of a true solid pulmonary lesion, with the exception of 22 false negative findings (60 false positive findings). Based on DTS images, readers incorrectly classified 10 pseudolesions (Figure 3) and 2 pleural plaques (10 mm and 12 mm in diameter) appearing as nodules or opacities on both CXR and DTS, in which they did not classify confidently whether a lesion was pulmonary or extra-pulmonary, and 8 benign lung nodules (7–10 mm in diameter) which were close to the anterior or posterior thoracic wall and were scored as indeterminate on DTS and, consequently, recorded as false negative findings.

Table 2 reports the different values of diagnostic performance and confidence for CXR and DTS. DTS showed a significantly (*p* < 0.05) increased sensitivity, specificity, and overall diagnostic accuracy and area under ROC curve compared to CXR alone. Mean effective dose was 0.06 mSv (range 0.03–0.1 mSv) for CXR, 0.107 mSv (range 0.094–0.12 mSv) for DTS, and 3 mSv (range 2–4 mSv) for CT (CXR vs DTS vs CT; *p* < 0.05).

## 4. Discussion

In this study, DTS was found to have a fairly good sensitivity and specificity in the detection and correct interpretation of lesions identified on CXR in areas of the chest difficult to assess on traditional radiography for the presence of overlying structures and poor penetration by the X-ray beam. In particular, DTS showed a better assessment and interpretation of lesions in the paradiaphragmatic zones and in the apical lung zone than in the hilar and retrocardiac lung regions. In most patients, DTS allowed interpretation of the lesions as extra-pulmonary or as pseudolesions, reserving CT for 108 patients (31%).

Digital tomosynthesis is a tomographic imaging technique that allows reduction of the visual distraction from overlying anatomical structures. This is particularly useful to assess regions where the anatomical noise results in poor conspicuity of the lesions, including the lung apices, lung bases, and the central areas of the lungs adjacent to the vessels. In a phantom study by Kim et al. DTS was compared to CXR and dual-energy subtraction (DES) for the detection of lung nodules of variable size in different locations [26]. The metric used was the lesion localization fraction (LLF), which is the number of lesion localization divided by the total number of lesions. DTS was superior to both CXR and DES for the detection of nodules of any size in the paramediastinal region, while it was not superior to CXR and DES for the detection of nodules of any size in the retrodiaphragmatic region. It was superior to CXR and DES in the apical and lateral pulmonary region but only for small nodules, otherwise the results were similar to those of CXRs.

In our study, DTS showed a good performance in the evaluation of lung bases. DTS may have limitations in the assessment of this zone mainly due to the limited depth resolution related to its geometric frontal plane acquisition that can result into misinterpretations of lung lesions located in the proximity of the chest wall and in the lung bases [27]. Langer et al. reported a higher sensitivity of DTS compared to CXRs for nodules abutting the pleura and those located within the periphery, even if not as high as expected due the presence of motion artifacts and abdominal structures, which may cause blurring of lesions in the immediate subdiaphragmatic zone [28]. In our study, the better results of DTS in this zone was probably also related to the exclusion from our analysis of DTS images with significant breathing artefacts impairing a correct assessment of the exam. DTS was shown to be useful for the evaluation of the pulmonary apical regions. This region is obscured on traditional X-rays by overlying bony structures (ribs, clavicle, and scapula) but pulmonary vasculature is scant and, therefore, DTS could maximize the detection rate of lesions by limiting soft tissue clustering. Hilar regions remain challenging areas to be evaluated on both CXR and DTS. As the acquisition of DTS images is limited to a restricted angle, the isotropic resolution of modern CT cannot be achieved by DTS and this prevents the complete removal of superimposed tissues [29]. Galea et al. suggested that the detection of hilar lesions is not improved with DTS; however, DTS is more specific and increases the inter-reader agreement compared to CXR [30]. Langer had similar results, reporting that sensitivity near the hilum was not ideal for DTS nor for CXR, even if only a small number of hilar lesions were assessed [28].

In our study, DTS was able to confirm or rule-out the vast majority of pulmonary lesions and to differentiate true lung opacities from those due to pleural or thoracic wall lesions or pulmonary pseudo-lesions. This resulted in an improved diagnostic accuracy and confidence, with only a modest increase in the radiation dose and interpretation time compared to CXR. The results of our study confirm the results reported in literature [9,10,11,12,13,14,16], which showed that DTS can rule out pseudolesions and overt benign lesions in the general patient population, and recommend CT only for patients with true potentially malignant parenchymal lesions. The main advantages of DTS is to solve doubtful findings directly in the X-ray unit without moving the patient to the CT, with an effective dose comparable to CXR and a lower radiation-dose than CT [31]. DTS can be scheduled immediately after CXR (e.g., the same day or few days after CXR), with comparable examination times. In our series, DTS resulted in a significant reduction in CT use, since DTS identified doubtful CXR findings as definite or probable benign pulmonary or extra-pulmonary lesion, or pulmonary pseudo-lesion deserving no further diagnostic work-up in 245/353 (69%) of patients, with only 22 (6%) false negatives. According to these results, DTS could be easily introduced in the routine diagnostic work-flow as a case-solving technique in suspected or equivocal pulmonary lesions on CXR, especially in regions of the chest in which lesions are most often missed or misinterpreted [4], reserving a CT examination only for patients with suspicious or indeterminate findings, with a consequent optimization of CT resources [10,32]. These results were proven only in solid nodules, pulmonary opacities, or pleural plaques, since we did not observe any ground-glass nodules, as DTS may have some limitations in the detection of sub-solid nodules [33].

The principal limitation of our study was its retrospective nature; the second limitation was the consensual analysis of the CXR and DTS images without assessment of the inter-reader variability; and the third limitation was the presence of multiple reference standards including CXR for pseudolesions or overt benign lesions, and CT or histology for positive pulmonary lesions.

To the best of our knowledge, this is the first study investigating the capability of DTS as a problem-solving imaging technique for patients with suspected pulmonary lesions in the so-called hidden areas on CXR.

## 5. Conclusions

In conclusion, Digital Tomosynthesis allowed the confirmation or exclusion of the presence of true pulmonary lesions in the hidden areas of the chest in about two-thirds of the patients.

## Figures and Tables

**Figure 1 diagnostics-13-01010-f001:**
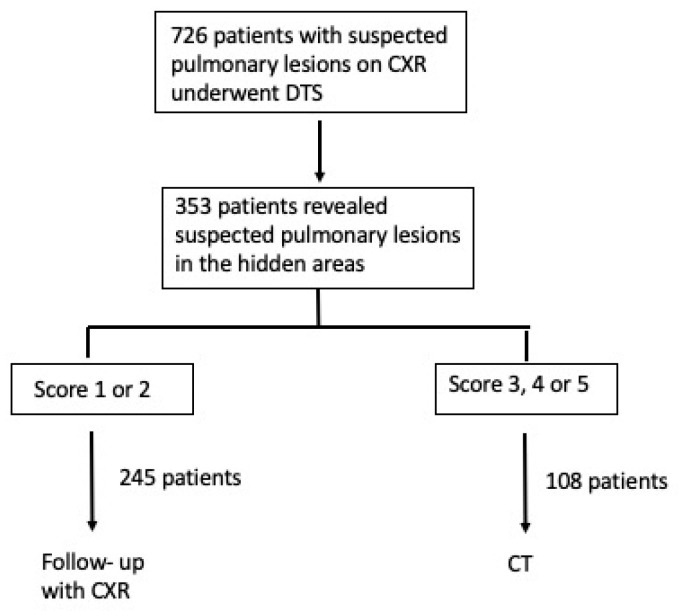
Flow diagram of the study population.

**Figure 2 diagnostics-13-01010-f002:**
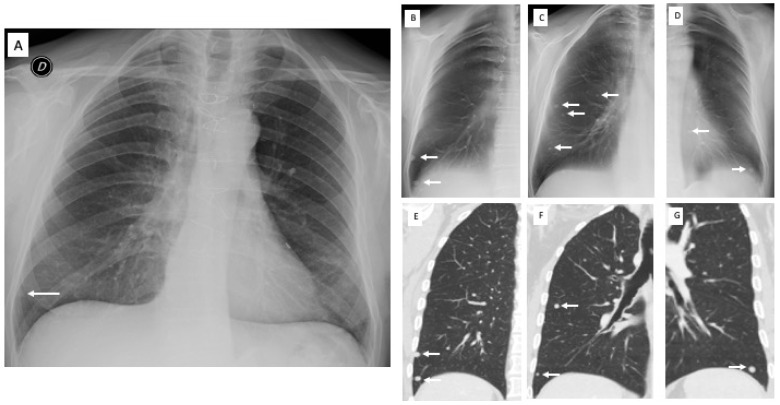
Pre-operative X-ray for colorectal carcinoma of a 56-year-old man. (**A**) Posteroanterior chest radiography in the upright position shows one suspected pulmonary nodule in the right lung base (arrow). (**B**–**D**) Digital tomosynthesis images confirm the previous nodule and identifies additional nodules (arrows) in both lungs, subsequently confirmed by CT (**E**–**G**).

**Figure 3 diagnostics-13-01010-f003:**
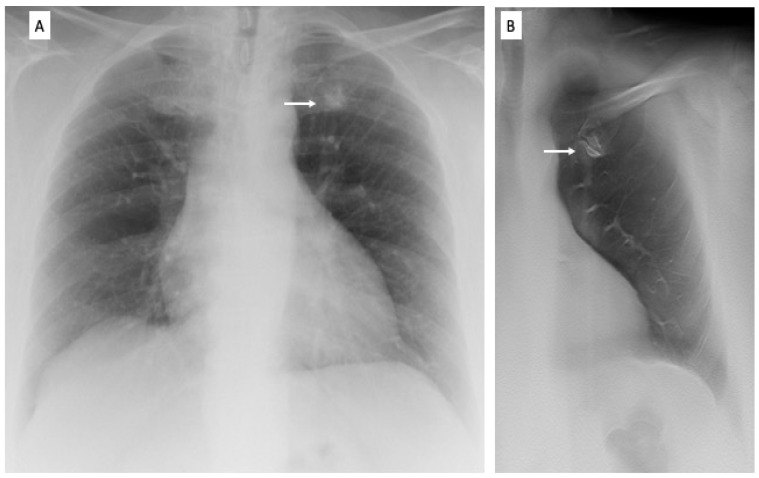
Pre-operative X-ray of a 72-year-old woman for skin melanoma. (**A**) Posteroanterior chest radiography in the upright position shows one suspected pulmonary nodule in the left apex (arrow). (**B**) Digital tomosynthesis image clarifies that the same opacity corresponds to a costal arthrosic hypertrophy of the anterior arch of the first left rib (arrow). Readers provided a confidence score of 1.

**Table 1 diagnostics-13-01010-t001:** Radiologic patterns and final diagnoses of all DTS findings.

Diagnoses	*n*	Mean Size (cm) ± SD	Size Range (cm)
**Pulmonary opacities**			
Non-tumoral lung opacities	32	2.3 ± 0.3	0.5–3
Squamous cell carcinomas	5	2.5 ± 0.5	2–3
Benign lung nodules	28	1.1 ± 0.3	0.5–1.5
Peripheral adenocarcinomas (nodules)	5	2 ± 0.7	0.5–1.5
Lung metastases	7	1.6 ± 0.7	1–1.8
**Pulmonary scars (#)**	26	1.1 ± 0.3	0.5–1.5
**Pleural plaques**	12	2.4 ± 0.6	1–3
**Pulmonary pseudolesions**			
Areas of increased opacity	92	-	-
Vascular kinking or ectasia	15	-	-
Auricula or mediastinal profiles	10	-	-
Lung variants	5	-	-
**Total**	237	2.3 ± 1.1	0.5–3

**Table 2 diagnostics-13-01010-t002:** Diagnostic performance. (*): Due to a sensitivity and specificity <50%, positive and negative likelihood ratios were not calculated.

	CXR	95% CI	DTS	95% CI	*p*
Sensitivity (%)	15 (16/103)	9.15–24	92 (95/103)	85.27–96.59	0.0001
Specificity (%)	9 (13/134)	5.27–16.02	91 (122/134)	84.88–95.29	0.0001
Positive likelihood ratio	(*)	(*)	10.3 (95/91)	5.99 to 17.72	
Negative likelihood ratio	(*)	(*)	0.09 (8/91)	0.04 to 0.17	
PPV (%)	12 (16/137)	6.82–18.27	88 (95/107)	81.23–94.07	0.0001
NPV (%)	13 (13/100)	7.11–21.2	93 (122/130)	88.23–97.31	0.0001
Accuracy (%)	12 (29/237)	8.35–17.09	91 (217/237)	87.26–94.76	0.0001

## Data Availability

The data presented in this study are available on request from the corresponding author.

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
