# Peer review of "Digital Tomosynthesis as a Problem-Solving Technique to Confirm or Exclude Pulmonary Lesions in Hidden Areas of the Chest"

_diagnostics, 2023, doi:10.3390/diagnostics13061010_

Round 1

Reviewer 1 Report

This article describes the effectiveness of digital tomosynthesis (DTS) on lung nodules.

The study is interesting in that the DTS is utilized in the diagnosis of lung nodules in hidden area, which is available at a relatively lower cost and under lower radiation dose.

However, there are some questions to be solved.

Major:

1)     The criteria of sensitivity, specificity, positive likelihood ratio, and negative likelihood ratio should be fully elucidated.

2)     The authors should explain the way how they defined that the relevant cases are negative for lung nodules.

3)     The uniqueness of the current study should be more elucidated, in comparison to the previous studies.

Author Response

Please, find attached the response to the reviewer's comments.

Reviewer 2 Report

Dear Authors,

this is a well-written manuscript presenting a very interesting diagnostic issue, which is the diagnostic approach of pulmonary lesions in hidden areas of the chest. Please pay attention to the following comments and queries pertaining to your manuscript:

1.      Figure 1: please change the subtitle us such: Flow diagram of the study population.

2.      Figure 1: please correct us such: 108 patients (present: pstients).

3.      First point of major concern: you have used the same scoring system (1-5) in order to describe 3 different diagnostic values which are: 1. the intra- oder extrapulmonary position of a lesion (lines 107-112), 2.  the malignant character of a lesion (lines 116-136) and 3. the likelihood of true/false positive or true/false negative findings (lines 137-147). This may become very confusing for your audience. Please provide further explanation. Is this one scoring system containing three parameters or three different scoring systems using the same symbolism (which is actually incorrect).

4.      Second point of major concern: the table 1 presents several diagnoses of the DTS findings including histologic features (squamous cell carcinomas, benign lung nodules, peripheral adenocarcinomas etc.). This may become very confusing for your audience because of the impression that the radiological method has the ability to differ between histological characters. Later in the text (lines 186-202) becomes clear that the histological characterization of the figures has been done after surgery or biopsy. Please change the structure of your results’ presentation in order to achieve accuracy.

With Best Regards

Author Response

(The authors gave the same response as above.)

Round 2

Reviewer 1 Report

The manuscript has been well revised. The authors have responded to the comments appropriately.

Reviewer 2 Report

Dear Authors,

thank you very much for providing comprehensive and convincing answers to my questions and queries and accordingly revised your manuscript.

Best Regards